# Luminous Self-Assembled Fibers of Azopyridines and Quantum Dots Enabled by Synergy of Halogen Bond and Alkyl Chain Interactions

**DOI:** 10.3390/molecules27238165

**Published:** 2022-11-23

**Authors:** Ying Pan, Lulu Xue, Yinjie Chen, Yingjie Hu, Zhicheng Sun, Lixin Mo, Luhai Li, Haifeng Yu

**Affiliations:** 1Beijing Engineering Research Center of Printed Electronics Institution, Beijing Institute of Graphic Communication, Beijing 102600, China; 2Nanjing Key Laboratory of Advanced Functional Materials, Nanjing Xiaozhuang University, Nanjing 211171, China; 3School of Materials Science and Engineering, Peking University, Beijing 100871, China

**Keywords:** azopyridines, halogen bond, quantum dots, luminous self-assembled fibers

## Abstract

Herein, a simple approach for the fabrication of luminous self-assembled fibers based on halogen-bonded azopyridine complexes and oleic acid-modified quantum dots (QDs) is reported. The QDs uniformly align on the edge of the self-assembled fibers through the formation of van der Waals force between the alkyl chain of oleic acid on the QD surface and the alkyl chain of the halogen-bonded complexes, 15Br or 15I. Furthermore, the intermolecular interaction mechanism was elucidated by using Fourier-transform infrared spectroscopy (FTIR), Raman spectroscopy, and density functional theory (DFT) calculations. This approach results in retention of the fluorescence properties of the QDs in the fibers. In addition, the bromine-bonded fibers can be assembled into tailored directional fibers upon evaporation of the solvent (tetrahydrofuran) when using capillaries via the capillary force. Interestingly, the mesogenic properties of the halogen-bonded complexes are preserved in the easily prepared halogen-bonded fluorescent fibers; this provides new insight into the design of functional self-assembly materials.

## 1. Introduction

The design and synthesis of molecules that can self-assemble into functional supramolecular structures with fascinating properties through multiple noncovalent interactions is a frontier in the material and chemical research fields. Several organic self-assembled supramolecular functional fibers, which show promise for biomedical applications, have been synthesized via electrospinning and inter- or intramolecular interactions, resulting in well-defined morphologies [1,2,3,4,5,6,7,8]. The halogen bond, a versatile noncovalent interaction that has directionality, tunable interaction strength, good hydrophobicity, and is compatible with the large size of halogen atoms, has been used for the fabrication of supramolecular assemblies, such as photoresponsive mesogens [9], supramolecular gels [10,11,12], microphase structures [13], and photo-actuators [14]. However, its application to supramolecular functional fibers has been little explored. The first report of halogen-bonded fibers, published in 2013, involved the use of bis (pyridyl urea) and 1,4-diiodotetrafluorobenzene in polar media to achieve gelation [10]. Subsequently, the analogous halogen bond-based supramolecular sol-gel transition of azopyridines was reported; however, halogen-bonded fiber structures were not discussed [11,12].

Azopyridines, which exhibit interesting self-assembly and photoresponsive abilities, have been used to fabricate supramolecular assemblies and are regarded as the most widely used components in the field of supramolecular chemistry [15,16]. The majority of reported azopyridine self-assembled fibers are constructed through hydrogen bonds and ionic bonds, and their self-assembly conditions and morphologies have been established. However, further investigation is required to develop supramolecular fibers with optical or electrical functions by mixing them with inorganic nanomaterial components. Quantum dots (QDs) are zero-dimensional inorganic materials with unique features, including high fluorescence efficiency, a narrow emission band, and tunable emission, owing to their size dependence; thus, they are commonly employed in solar cells, emitting diodes, biomedical imaging, and fluorescent anti-counterfeiting technology [17,18,19,20,21,22,23,24,25]. Recently, electrospun fluorescent fibers containing organic components and QD composites, such as CdSe/ZnS core-shell QDs and CdSe QDs, have been realized for use in polymeric lasers and optical sensors [26,27]. Furthermore, an intense circularly polarized luminescent material was prepared by forming a novel luminescent chiral nanotube using a chiral lipid gelator as a chiral template for the QDs [28]. However, the formation of azopyridine supramolecular fluorescent or directional fibers that incorporate inorganic nanomaterials through combined halogen bonds and van der Waals force has not yet been reported.

Considering this, we used our previously reported azopyridine halogen-bonded liquid crystal (LC) complexes (nX, X = Br, or I) to assemble fibers that exhibit an ordered orientation and luminescence (Figure 1); details of the synthesis are presented in the ESI. We established that 15Br could promote the targeted directional arrangement of disordered supramolecular fibers with the aid of a capillary when tetrahydrofuran (THF) is used as a solvent. Mixing oleic acid-modified CdSe/ZnS core-shell QDs with the LC complexes resulted in the formation of large luminous fiber crystals of 15X@QDs (X = Br or I), in which the QDs were aligned along the edge of the self-assembled fibers due to van der Waals force between the long alkyl chain of oleic acid on the QD surface and 15Br or 15I. This is a novel approach for the fabrication of supramolecular structures with new properties and functions that can be applied in drug detection, biosensors, and electroluminescent devices.

## 2. Results and Discussion

The halogen-bonded azopyridine fibers spontaneously formed either in the THF or upon the evaporation of one drop of the THF solution on the surface of the glass substrates in random order. The optical and scanning electron microscopy (SEM) images of the fibers (Figure 1a,b, respectively) reveal that the width of the supramolecular fibers of 15Br is on the micron scale. The effect of the alkyl chain length of the halogen-bonded complexes on the morphology of the fibers was investigated. Self-assembled fibers were formed in the THF using bromine-bonded complexes with alkyl chain lengths of 7 to 15 carbons (Appendix A, ESI). The fibers of 15Br had a large aspect ratio, while the corresponding azopyridine derivatives did not form fibrous structures, which suggests that halogen bonds play a key role in facilitating the formation of supramolecular fibers.

In addition, taking 15Br as a typical example, an increase in the mass concentration of 15Br in the THF from 0.1 to 1.0 wt% resulted in an increase in the number of visible self-assembled fibers (Appendix A, ESI). Subsequently, the morphologies of the fibers obtained from 15Br and its pristine counterpart, A15AzPy, were studied using six organic solvents with different polarity indices. The nature of the solvent influenced the morphology of the self-assembled fibers, indicating that the morphologies of the fabricated halogen-bonded fibers depended on the nature of the chosen organic solvent (Appendix A, ESI).

LCs are mesophasic, which means they exist between the melting and clearing points. Below its melting point, the material forms a normally ordered solid, and above the clearing point, it forms an isotropic liquid. The thermal properties of the self-assembled fibers fabricated by recrystallization of LC molecular 15Br were studied using differential scanning calorimetry (Appendix A). The crystal-to-mesophase transition and the mesophase-to-isotropic liquid transition temperatures increased significantly from 100.2 °C to 157.6 °C for the 15Br and 143.6 °C to 160.6 °C for the fibers, respectively. Moreover, the 15Br fibers have a narrower LC range than 15Br. In addition to obtaining optical microscopy images, the crystals of 15Br were purified in the THF and exhibited high melting and clearing points. These results were supported by the powder X-ray diffraction results (Appendix A). Three diffraction peaks appear in the low-angle region for the 15Br fibers, similar to those observed for 15Br, and the *d*-spacing (1.78, 0.89, and 0.46 nm), which has a ratio of 1:1/2:1/4 and is consistent with a lamellar structure [29]. The peak at 1.78 nm, observed in the first-order reflection of the 15Br fibers, is sharp compared to the peak obtained for the pristine material, suggesting that the 15Br fibers are more ordered than those of the pristine material [30].

Moreover, 15Br could form directional self-assembled fibers in capillaries upon evaporation of the THF, owing to the capillary force [31,32,33], as shown in Figure 1c. This is a key technology in the development of high-performance organic materials.

Supramolecular luminescent fibers are particularly important because of their potential applications in interdisciplinary research, such as light-emitting electrochemical cells [34] and diagnostic devices [35,36]. The general method for the production of luminescent fibers involves electrospinning QDs in a polymer solution; however, this process has low production efficiency because the polymer solution has a slow rate of reaction, and the procedure is labor intensive [1]. In this study, novel luminescent halogen-bonded fibers were designed and easily obtained by mixing 15Br and oleic acid-modified CdSe/ZnS QDs in THF. As shown in Figure 1d,e the majority of the self-assembled fibers with bright fluorescence can be observed at the edges of the structures after the addition of QDs. The QDs supply fascinating luminescence properties without disturbing the fiber structures. Surface tension can cause the fluid to flow rapidly over the surface and remain almost stagnant in the internal area, while the QDs are carried to the edges by the Marangoni flow [37,38]. Furthermore, by increasing the volume-to-volume ratio of the QDs from 3:1 to 1:1, the aspect ratio of the self-assembled organic/inorganic hybrid composite (15Br@QDs) fibers can be increased, as shown by laser scanning confocal microscopy (Appendix A). To eliminate the effects of the solvent used for the QDs on the morphology of the fibers, *n*-hexane and the QDs in *n*-hexane were added separately to the THF solution of 15Br, using the same ratio as used previously, and analyzed by SEM and energy-dispersive X-ray spectroscopy (EDS). The mixture of 15Br in THF and the oleic acid-modified CdSe/ZnS QDs in *n*-hexane spontaneously self-assembled into fibers with a high aspect ratio (Appendix A), while inhomogeneous structures self-assembled upon the addition of *n*-hexane to 15Br in the absence of QDs (Appendix A). The EDS results indicate that QDs were present on the fibers of the 15Br@QDs (Appendix A), which contributed to the formation of large-aspect-ratio self-assembled luminous fibers. In addition, A15AzPy@QDs could also form self-assembled fibers due to the van der Waals interactions between alkyl chains (Appendix A). The fluorescence properties of 15Br@QDs in solution are shown in Figure 2a. The emission peaks of CdSe/ZnS QDs exhibited a similar pattern to those of the resultant fibers, while being slightly red-shifted, indicating the conservation of the fundamental fluorescence properties of the QD-assisted luminescent fibers.

Raman spectroscopy is a powerful tool for investigating halogen bonds [39,40]. As shown in Figure 2b, the Br–Br stretching peaks of the 15Br fibers shifted to a lower wavenumber relative to those of 15Br (219.7 cm^−1^ to 207.2 cm^−1^), which indicates not only that the 15Br fibers are halogen bonded, but also that the recrystallization of the 15Br fibers from 15Br in THF weakens the halogen bond and leads to a decrease in the Br–Br vibration frequency of the 15Br fibers. Furthermore, the Br–Br stretching vibration peaks of 15Br@QDs observed at 216.8 cm^−1^, suggest that the weak van der Waals force between alkyl chains has a subtle influence on the halogen bond. This suggests that the halogen bond plays a major role in the self-assembly of the azopyridine fibers, and the QDs further aid the self-assembly process via van der Waals force, leading to the formation of luminous fibers.

Figure 2c shows the UV-vis absorption spectra of 15Br in the THF, CdSe/ZnS QDs, 15Br@QDs solution, and 15Br@QDs film. The maximum absorption of 15Br and 15Br@QDs in THF occurs at wavelengths of 359 and 354 nm, respectively, which is attributable to the azobenzene π–π * bands. The spectrum of 15Br@QDs reveals a small blue shift relative to the 15Br spectrum, which is caused by the substituent effect of the carboxylate groups of oleic acid on the surface of the QDs and bromine atom. In contrast, the peak at 363 nm in the spectrum of 15Br@QDs in solution was red-shifted in the spectrum of the 15Br@QDs film. This red shift may be attributable to the *J*-aggregation of the chromophores, which are partly arranged in close proximity to each other and with coplanar transition dipoles [41,42,43,44,45,46,47].

The distribution of CdSe/ZnS QDs in the halogen-bonded fibers was observed using transmission electron microscopy (Figure 3). Interestingly, the 15Br fibers can act as a template for the arrangement of QDs during self-assembly. Thus, the CdSe/ZnS QDs were uniformly distributed along the edge of the halogen-bonded fibers owing to the van der Waals force between 15Br and the QDs, resulting in the preservation of the original fluorescence properties of the QDs, as confirmed by the absorption and fluorescence spectral data. In addition, the EDS maps of the fibers indicate the presence and even distribution of Br, S, Cd, Se, and Zn in the QDs on the fibers (Figure 3c–g). The introduction of long-alkyl-chain-functionalized QDs promotes interactions between the QDs and halogen-bonded complexes, resulting in luminous fibers.

Furthermore, the LC properties of the 15Br@QDs were determined using polarizing optical microscopy; 15Br@QDs were shown to have a focal conical fan texture similar to that of 15Br (Appendix A). Moreover, the aligned 15Br@QDs mesogens show strong anisotropy in their polarized UV-vis absorption spectra, and the orientation factor (0.052) compares favorably with that of pristine 15Br (0.004) (Appendix A).

To evaluate the bonding capability of the AzPy derivatives and QDs as a function of the halogen atom, molecular iodine was used as a Lewis base instead of molecular bromine for the preparation of iodine-bonded 15I@QDs complexes. The aspect ratio of the self-assembled luminous 15I@QD fibers is expected to be large compared to that of the 15Br@QDs owing to the presence of the QDs (Figure 4a,b). This could be because the halogen bonds decrease in strength in the following order: I > Br > Cl [48,49,50,51], and the relatively weak bromine bonds led to a reduction in the aspect ratios. The standard deviations of the aspect ratios of the 15Br, 15Br@QDs, and 15I@QDs fibers are calculated in the ESI.

Considering the photoresponsivity of our prototype molecule 15I, the photoactivity of the 15I@QDs was monitored before and after UV irradiation at 360 nm in the THF solution. Exposure of the 15I@QD solution to UV light resulted in a gradual decrease in the absorption peak at 358 nm, which can be attributed to a π–π * transition, and a gradual increase in the peak at 442 nm, which can be attributed to the n–π * transition (Figure 4c). This is the result of photoisomerization of AzPy molecules from their *trans* to *cis* isomers. The intensity of the absorption peak at 358 nm gradually increased, and that at 442 nm decreased when the irradiated sample was kept in the dark, indicating that the 15I@QDs underwent trans to cis to trans isomerization (Figure 4d). However, this photochemical phase transition was not observed for 15Br@QDs, which produced a spectrum analogous to that of the brominated 15Br compound when irradiated with UV light.

Density functional theory (DFT), specifically the Gaussian 16 program, was used to investigate possible 15X@QDs (X = Br or I) interactions [52]. Geometry optimization calculations were performed using the B3LYP DFT-D3 method [53,54,55], and the 6−311G + (2d,2p) basis set was used to locate all stationary points involved. The vibrational frequencies were computed at the same level of theory to check whether each optimized structure is an energy minimum or transition state and to evaluate its zero-point energy. The binding energy (E_b_) is defined as E_b_ = E_A + B_—(E_A_ + E_B_), where E_A + B_ is the total energy of A and B combined, and E_A_ + E_B_ is the sum of the total energies of A and B before the combination. A and B refer to 15X and alkyl chains of oleic acid on the QD surface, respectively.

The E_b_ energies of 15Br@QDs with 4, 8, 15 overlap of alkyl chains via van der Waals force were calculated to be −0.15125 eV, −0.30105 eV and −0.52469 eV, respectively, and the E_b_ of 15I@QDs were calculated to be −0.15254 eV, −0.30102 eV and −0.52463 eV, respectively (Figure 5). The energy analysis shows that the absolute value of binding energy of 15X@QDs is getting bigger and bigger as the overlap of alkyl chains increases between QDs and 15X molecules via van der Waals force, which led to the further stabilization of the entire system of 15X@QDs.

## 3. Materials and Methods

Materials: 4-aminopyridine, K_2_CO_3_, acetone and other chemical reagents were obtained from Sigma-Aldrich, Saint Louis, MO, USA. CdSe/ZnS QDs were purchased from Wuhan Jiayuan Quantum Dot Co., Ltd. Wuhan, China. The maximum emission wavelength is 625 nm ± 5 nm, and the size is 5–10 nm. The specification is 30 mg powder QDs dispersed in 10 mL *n*-hexane solvent.

Characterizations: ^1^H NMR spectra were executed on a Bruker Avance III 400. Differential scanning calorimetry (DSC) examination was performed on a Perkin-Elmer DSC 8000 with a heating and cooling rate of 10 °C/min. The morphologies of fibers were observed using scanning electron microscopy (SEM, Zeiss EVO18, Oberkochen, BW, Germany), polarizing optical microscope (POM, LEICA DM2700 M, Wetzlar, Hesse, Germany) and transmission electron microscopy (TEM, Tecnai G2 F20, Hillsboro, OR, USA). The powder X-ray diffraction analysis (XRD) was implemented on a Philips X pert pro. The FT-IR analysis and the UV-vis analysis were measured using Nicolet 510P IR spectra and a UV/VIS/NIR spectrometer (Perkin-Elmer lambda 950, Waltham, MA, USA). Laser scanning confocal microscopy was recorded on the Zeiss LSM800.

Simulation method: All calculations were performed using the Gaussian 16 program. Geometry optimization calculations were performed using the B3LYP DFT-D3 method.

## 4. Conclusions

In conclusion, we report that the self-assembled fibers can be developed from halogen-bonded complexes by varying the alkyl-chain lengths, concentrations, and solvents. We found that 15Br, with the aid of a capillary, can form directional self-assembly fibers in THF. Interestingly, the self-assembled luminous fibers were obtained by the synergy interactions of halogen bonds and van der Waals force between the oleic acid groups of the QDs surface and the alkyl chain of the halogen-bonded complexes, which is critical to potential applications, such as drug detection, biosensors, electroluminescent devices, and other novel optical devices.

## Data Availability

Not applicable.

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
