# Peer review of "Luminous Self-Assembled Fibers of Azopyridines and Quantum Dots Enabled by Synergy of Halogen Bond and Alkyl Chain Interactions"

_molecules, 2022, doi:10.3390/molecules27238165_

Round 1

Reviewer 1 Report (Previous Reviewer 1)

The manuscript describes formation of luminous fibers of azopyridines and quantum dots. The results describing formation and properties of these fibers  are interesting and deserve publication. There are several issues with the interpretation of the intermolecular interactions which should be addressed before manuscript can be published.

The author suggested that formation of the aggregates are related to the vdW interaction between alkyl chains (Fig 5) which does not actually involve dihalogens. However, this contradict their earlier suggestion. .      

First, based on the vibrational spectra, the authors claim that the Br–Br stretching vibration shifts from 207.2 cm-1 to 147.2 cm-1 after combining 15Br with the QDs suggests that the recrystallized 15Br interacts with the long alkyl chain of the QDs via Van der Waals forces. From the presented in Fig 2 spectra, it seems that the very weak band at 147 cm-1 is not related to Br-Br stretching because similar weak peaks are observed in the 15 Br and 15Br fibers. And in any case, contrary to the authors suggestion, the weak vdW forces would not result in such a large shift of Br-Br vibration, and this vibration is related to Br-Br covalent bond, not  halogen bond between Br2 and azopyridine.    

Second, if weak vdW forces are responsible for the formation of fibers, it is not clear why the replacement of Br2 with I2  (which are not involved in bonding) would affect properties of fibers.   

Third, the authors suggested that irradiation of 15I@QD results in a shift from p-p* to n-p*   transition. However, they did not specify what this  n-p*  transition represent and if it is trans-cis isomerization why it would happen with iodine but not with bromine-bonded azopyridines. .  

Fourth, in Fig 5,  the authors showed optimized structures. They did not write if they check other possible arrangements, since there could be several  local minima for interaction of large molecules. It is not surprising that dispersion interaction between such chain would stabilize the aggregate. However,  the halogens are not involved in the interaction at all, so they would not substantially affect properties of the aggregates. Also, the authors should be more careful with the description of the negative values, since the statement  “the energy of the entire system of 15X@QDs (EA+B) is greater than the sum of QDs 232 and 15X molecules” could imply that the aggregate is less stable and Eb  would be positive, not negative (as it is).  

Overall, I think that the manuscript presents interesting self-assembled fibers. However, there is no real explanation why halogen bond would affect their formation, and no real elucidation of the intermolecular interaction mechanism (as it is claimed in the Abstract).   

Author Response

See the attachment.

Reviewer 2 Report (Previous Reviewer 2)

The authors have satisfactorily addressed the reviewer comments, and hence may be accepted for publication.

Author Response

Thanks for your review. 

Reviewer 3 Report (New Reviewer)

The authors reported a simple approach for the fabrication of luminous self-assembled fibres based on halogen-bonded azopyridine complexes and oleic acid-modified quantum dots. The topic is interesting; however, I have some comments.

1.      The sentence “the bromine-bonded fibres can be assembled into 20 tailored directional fibres” in the abstract is confusing,

2.      “Van der Waals force” should be “van der Waals force”. Such as “a Van der Waals' force”; “Van der Waals force interactions” should be corrected.

3.      “between the long alkyl chain on the QD surface and 15Br or 15I.” is confusing. The long alkyl chain should be the chain of oleic acid.

4.      After mixing 15Br and oleic acid-modified CdSe/ZnS QDs together, the N…Br halogen bond might be destroyed partly. A15AzPy molecules might form the N…H hydrogen bond with oleic acid. In common, the N…H hydrogen bond is stronger than the N…Br halogen bond. The QDs uniformly align on the edge of the self-assembled fibres of 15B, indicating the van der Waals interactions between the side chain of 15B and oleic acid are stronger than the hydrogen bonding. For 15I, nanoparticles locate at the center of the fiber structures. I suggest that one possible reason may be that the intermolecular I…O and I…H bonds are stronger than I…N bonds.

Please see the references: The Journal of Physical Chemistry C 2022, 126 (12), 5777; Applied Surface Science 2018, 433, 1075; The Journal of Physical Chemistry Letters 2016, 7 (16), 3164.

Thus, the DFT calculation should be considered again,

5.      “and 143.6 °C to 160.6 °C the fibers,” it is a wrong description.

6.      “weak bromine bonds cause an agglomeration of nanoparticles at the center of the fiber structures” for 15Br, the nanoparticles should at the edge of the fiber. So the sentence is confusing.

7.      In Figure 2b, 15Br is solution or solid.

8.      The morphology of 15Br fibre and luminous self-assembled fibre is different. Except for the interchain van der Waals interactions are considered, the interactions between Br/I and -COOH have to be considered.

Author Response

See the attachment.

Reviewer 4 Report (New Reviewer)

Dear Editor,

I have investigated the manuscript and my comments are listed below.

The manuscript is well written and designed and includes worthy results.

I recommend that this manuscript can be accepted in this present forms to be publish in Molecules, along with the small corrections.

- Figure 3 (f) and (g) is not clear.

- In Figure 5, the authros should present the optimized structure with the adding Br and I.

Round 2

Reviewer 1 Report (Previous Reviewer 1)

The authors addressed the most apparent issues and the part of the manuscript describing the preparation of the luminous fibres is sound. Therefore,  the manuscript could be published despite the tenuous mechanistic interpretations.        

Reviewer 3 Report (New Reviewer)

It can be published in present version.

This manuscript is a resubmission of an earlier submission. The following is a list of the peer review reports and author responses from that submission.

Round 1

Reviewer 1 Report

The manuscript presents preparation and characterization of luminous fibres of azopyiridines  and quantum dots.  It can be published eventually, but in the current form it cannot be recommended for publication. 

The manuscript is difficult to read. Besides very poor English, the authors did not provide necessary information and did not check the article. For example, they did not provide any data in the article (and very limited data in the SI) about preparation of azopyridine complexes with Br2  or I2.  They presented pairs of Laser scanning confocal microscopy images of fluorescent fibres  in Figure 1 and 4 without explanation of differences between them. X-axis in Figure 2b is in wavenumber, not wavelength, as written. When they discuss results of computations, they wrote equation for Eb but did not provided values of Eb. They wrote, however, that “interplay of hydrogen bonding and halogen bonding …. enhance the energy of the whole system slightly, which is 0.10121 eV and 0.25669 eV, respectively” . Does this mean that energy of the complex is higher than the energy of component (i.e., interaction is repulsive)? And   0.10121 eV and 0.25669 eV (2-6 kcal/mol) are not “slightly” if referred to intermolecular interactions.

There are many questions to the authors’ interpretation of their data. Figure 2 suggests that it is just quantum dots which are luminescent, and not fibres. Accordingly, it seems that term “luminescent fibres” is not correct.  Also, they suggested hydrogen bonding between 15D and QD. But Br2 and I2 can form two halogen bonds (on both sides).  Did they check if bromine or iodine form halogen bond with the acid? It seems that their data can be also explained by formation of another halogen bond between X2  and oxygen in the acid.        

Reviewer 2 Report

Please find the Reviewers comments pdf attached.

Round 2

Reviewer 1 Report

The revised version still requires extensive editing before it can be recommended for publication.The authors should  consider using some editing service to make their article readable (and clear enough for the appropriate review of possible scientific issues). Even sentences which are more or less consistent with English grammar, are still very ambiguous.    Let me illustrate this with the very first sentence in Results and Discussion.  "The halogen-bond azopyridine fibres spontaneously formed in THF or upon evaporation of one drop of the resultant solution on the surface of glass substrates in a random order." What "resultant solution" is? What "surface  of glass substrates" means? "Random order" of what? 

Also: how "the pristine material"  (supposedly 15B) was prepared?

A few more examples from the next page: What  low "prodfibreuction  
efficiency" or
  "rates of the polymer solution" mean?  What "volume percentage of the QDs" means?  S
entences which were added in revisions are also ambiguous.

Overall, the same conclusion as previously: the manuscript seems to present interesting results and it is potentially publishable. However, the presentation should be substantially improved (and I mean really substantially improved, not just a few sentences mentioned above). It still cannot be recommended for publication.

Reviewer 2 Report

The authors have satisfactorily addressed the comments and is suitable for publication.